# Comparative Efficacy and Safety of Glucagon-like Peptide-1 Receptor Agonists in Children and Adolescents with Obesity or Overweight: A Systematic Review and Network Meta-Analysis

**DOI:** 10.3390/ph17070828

**Published:** 2024-06-24

**Authors:** Ligang Liu, Hekai Shi, Yufei Shi, Anlin Wang, Nuojin Guo, Heqing Tao, Milap C. Nahata

**Affiliations:** 1Institute of Therapeutic Innovations and Outcomes (ITIO), College of Pharmacy, The Ohio State University, Columbus, OH 43210, USA; liu.10645@osu.edu; 2Department of Bariatric and Metabolic Surgery, Fudan University Affiliated Huadong Hospital, Shanghai 201203, China; 22211280026@m.fudan.edu.cn; 3Department of Clinical Pharmacy and Pharmacy Administration, School of Pharmacy, Fudan University, Shanghai 200437, China; yufei_shi@fudan.edu.cn; 4Department of Pharmacy, Beijing Chao-Yang Hospital, Capital Medical University, Beijing 100054, China; jordan0309@126.com; 5Department of Endocrinology, Shanghai East Hospital, Tongji University School of Medicine, Shanghai 200331, China; 2233148@tongji.edu.cn; 6Department of Gastroenterology, The First Affiliated Hospital of Guangzhou Medical University, Guangzhou Medical University, Guangzhou 510180, China; tao_heqing@bjmu.edu.cn; 7College of Medicine, The Ohio State University, Columbus, OH 43210, USA

**Keywords:** GLP-1RAs, obesity, adolescents, efficacy, safety, network meta-analysis, semaglutide

## Abstract

Four glucagon-like peptide-1 receptor agonists (GLP-1 RAs) have been used in children and adolescents with obesity or overweight. This network meta-analysis was conducted to compare the efficacy and safety of these regimens. Embase, PubMed, and Scopus were searched on March 2023 and updated in June 2024 for eligible randomized controlled trials (RCTs). The primary efficacy outcomes were mean difference in actual body weight, BMI (body mass index), BMI z score, and waist circumference. Safety outcomes included nausea, vomiting, diarrhea, abdominal pain, injection-site reaction, and hypoglycemia. Eleven RCTs with 953 participants were eligible. Semaglutide exhibited greater effects in reducing weight, BMI, and BMI z score versus the placebo. Semaglutide was associated with greater weight loss and BMI z score reduction in comparison with exenatide, liraglutide, and dulaglutide. Semaglutide also significantly decreased BMI than exenatide. None of the four GLP-1 RAs were associated with higher risks of diarrhea, headache, and abdominal pain versus the placebo. Liraglutide was more likely to cause nausea, vomiting, hypoglycemia, and injection-site reactions than the placebo. Liraglutide also had higher odds of causing injection-site reactions than other GLP-1 RAs. Semaglutide appeared to be the most effective and safe option among four GLP-1 RAs in children and adolescents with obesity or overweight.

## 1. Introduction

Approximately one-third of children and adolescents in the United States are diagnosed with overweight or obesity [1]. This condition is likely to persist into adulthood for about 80% of adolescents with obesity [2]. Obesity is strongly associated with cardiovascular disease, hypertension, diabetes, non-alcoholic fatty liver diseases, polycystic ovarian syndrome, sleep apnea, as well as musculoskeletal and mental health disorders [3,4].

Childhood obesity is potentially associated with several behavioral factors, including decreased physical activity, excessive screen time, inadequate sleep, and unhealthy eating habits [5]. Therefore, lifestyle intervention is recommended as the initial approach for managing childhood obesity [6,7,8]. Pharmacological treatments are considered as adjunctive therapies in cases where lifestyle modifications alone do not achieve the desired results.

Currently, medications approved for the treatment of obesity in children and adolescents include orlistat, phentermine, phentermine-topiramate, liraglutide, and semaglutide [9,10,11,12,13]. Liraglutide and semaglutide, as glucagon-like peptide-1 receptor agonists (GLP-1 RAs), enhance glucose-dependent insulin secretion, slow down gastric emptying, increase satiety, and reduce food intake [14]. The use of GLP-1 RAs has increased among pediatric patients since the approval of liraglutide and semaglutide [12]. Bariatric surgery is generally reserved for individuals with severe obesity and multiple comorbidities who have not responded to other interventions [15].

As part of the comprehensive treatment, pediatricians and other primary care professionals often consider prescribing medications as an adjunct to health behavior and lifestyle interventions for adolescents aged 12 years and older with obesity [16]. A thorough assessment of the effectiveness and safety of these medications in pediatric patients is necessary [17]. GLP-1 RAs differ in structure, pharmacokinetics, dosing intervals, and administration methods, which could significantly influence their efficacy and tolerability [18]. Published meta-analysis has primarily evaluated the safety and efficacy of exenatide and liraglutide in children and adolescents with obesity [19]. Recently, more GLP-1 RAs have been used in this demographic. Dulaglutide has demonstrated promising results in youths with overweight and type 2 diabetes [20]. Furthermore, semaglutide received FDA approval for the treatment of obesity in the pediatric population in January 2023 [12]. Nevertheless, the comparative efficacy and safety of liraglutide, semaglutide, dulaglutide, and exenatide remain unknown.

Therefore, this systematic review and network meta-analysis was performed to investigate the comparative efficacy and safety of the four GLP-1 RAs in children and adolescents with obesity or overweight. We hypothesized that semaglutide might offer better efficacy outcomes and similar safety profiles compared to other GLP-1 RAs. 

## 2. Results

### 2.1. Search Results and Characteristics of Studies

The search retrieved 923 studies. After removing duplications and screening titles and abstracts, we conducted a full-text review of published studies. Finally, eleven studies were included in our analysis (Figure 1) [20,21,22,23,24,25,26,27,28,29,30]. Ten trials were randomized double-blind, placebo-controlled trials [20,21,22,23,24,25,26,27,28,29], while one was a crossover study [30]. Three randomized controlled trials (RCTs) were conducted with a focus on individuals with overweight and type 2 diabetes [20,21,22], five studies included patients with obesity [23,24,25,26,27], and three trials enrolled participants with severe obesity [28,29,30]. Geographically, five of the included trials were multinational [20,21,22,26,27], one was conducted in Germany [23], four in the US [25,28,29,30], and one in Belgium [24]. In total, this network meta-analysis included 953 children and adolescents, with 154 participants receiving dulaglutide, 201 receiving semaglutide, 147 receiving exenatide, and 451 receiving liraglutide. Among the included trials, one compared dulaglutide vs. placebo [20], five studied liraglutide vs. placebo [21,22,23,24,25], four evaluated exenatide vs. placebo [26,28,29,30], and one investigated semaglutide vs. placebo [27]. 

Dulaglutide was started at 0.75 mg weekly and escalated to 1.5 mg weekly [20]. For liraglutide, one trial began at 0.3 mg daily and escalated gradually up to 1.8 mg [21], another commenced at 0.6 mg and escalated to 1.8 mg [22], another at 0.3 mg daily and escalated up to 3 mg [25], and two studies started at 0.6 mg daily and escalated up to 3 mg [23,24]. For exenatide, two trials maintained a consistent dose of 2 mg weekly [26,29], while two other studies began with 5 mcg twice daily and escalated to 10 mcg twice daily after 1 month [28,30]. Semaglutide was initiated at 0.25 mg weekly and gradually increased to 2.4 mg weekly [27]. 

The majority of the included studies had small sample sizes, ranging from 11 to 66 participants. However, four RCTs were relatively large, involving between 134 and 251 participants [20,22,24,27]. Seven studies had a duration ranging from 5 to 26 weeks, while four studies had a longer duration from 52 to 68 weeks [22,24,27,29]. The mean age of the participants ranged from 9.9 to 16 years, with the percentage of male participants varying from 18.2% to 50%. The mean body weight of the participants ranged from 71.5 to 124 kg, while their mean body mass index (BMI) ranged from 33.9 to 42.5 kg/m^2^. The BMI z-score spanned from 2.94 to 3.9. Nine of eleven studies (82%) included lifestyle intervention concurrently, whereas two studies did not because the primary purpose was to assess the safety and tolerability of liraglutide [23,25]. A summary of included studies appears in Table 1. 

The risks of bias score were categorized into low risk, some concerns, and high risk. In the assessment, four trials (36.4%) had bias arising from the randomization process, one trial (9.1%) had a high risk of bias due to deviations from intended interventions, and five trials (45.5%) had some concerns or high risk of bias due to missing outcome data. Study level and overall risk of bias assessments are summarized in Appendix A. 

Figure 2 and Appendix A show the network graphs of the main outcomes and subgroup analysis of primary outcomes. Estimates of pairwise meta-analyses of all outcomes are presented in Figure 3 and Appendix A. 

### 2.2. Primary Efficacy Outcomes

#### 2.2.1. Actual Weight Changes

In contrast to the placebo, semaglutide (DMD: −17.67 kg; 95% CI: −23.18, −12.37) and exenatide (DMD: −3.47 kg; 95% CI: −5.85, −1.38), but not liraglutide (DMD: −2.27 kg; 95% CI: −4.82, 0.05) or dulaglutide (DMD: 0.02; 95% CI: −3.49, 4.01) showed a greater decrease in weight. Semaglutide was also associated with greater weight loss than exenatide (DMD: −14.21; 95% CI: −19.81, −8.45), liraglutide (DMD: −15.39; 95% CI: −21.30, −9.47), and dulaglutide (DMD: −17.75; 95% CI: −24.06, −11.05), as shown in Figure 3A. The SUCRA ranking suggested that semaglutide had the highest probability of being ranked first, followed by exenatide, liraglutide, and dulaglutide (Table 2).

#### 2.2.2. Changes in BMI and BMI z Score

BMI was reported in eight studies that used liraglutide, exenatide, and semaglutide as the intervention [20,22,24,26,27,28,29,30], so dulaglutide was not included in the comparison of BMI. Semaglutide (DMD: −5.99 kg/m^2^; 95% CI: −9.36, −2.72) was likely to be more effective in BMI reduction than the placebo. We observed no benefit of exenatide (DMD: −1.25 kg/m^2^; 95% CI: −3.39, 0.16) or liraglutide (DMD: −1.59 kg/m^2^; 95% CI: −4.67, 1.51) over the placebo. Furthermore, semaglutide (DMD: −4.70 kg/m^2^; 95% CI: −8.33, −0.35) was more effective for BMI reduction compared to exenatide (Figure 3B). When ranking all the agents, semaglutide had the highest probability of being ranked first for BMI reduction, followed by liraglutide and exenatide (Table 2).

Semaglutide was associated with a significant reduction in the BMI z score compared with the placebo, with DMD values ranging between −1.54 and −0.46. However, liraglutide (DMD: −0.13; 95% CI: −0.40, 0.11), exenatide (DMD: −0.09; 95% CI: −0.56, 0.37), and dulaglutide (DMD: −0.01; 95% CI: −0.49, 0.49) did not significantly reduce the BMI z score versus placebo. Moreover, semaglutide significantly reduced the BMI z score relative to liraglutide (DMD: −0.87; 95% CI: −1.44, −0.27), exenatide (DMD: −0.91; 95% CI: −1.62, −0.12), and dulaglutide (MD: −0.99; 95% CI: −1.72, −0.25), as presented in Figure 3B. Ranking probability showed that semaglutide was ranked first at 98.94%, followed by liraglutide, exenatide, and dulaglutide (Table 2).

#### 2.2.3. Changes in Waist Circumference and Total Tissue Fat

Compared to the placebo, semaglutide (DMD: −12.01 95% CI: −23.70, 1.32), liraglutide (DMD: −3.02 95% CI: −16.55, 8.57), exenatide (DMD: −2.26 95% CI: −11.54, 6.73), and dulaglutide (DMD: −0.95 95% CI: −13.90, 11.94) failed to reduce waist circumference significantly. No significant difference was observed between the different GLP-1 RAs (Figure 3A). When ranking four GLP-1 RAs, semaglutide had the highest probability of being ranked first, followed by liraglutide, exenatide, and dulaglutide (Table 2).

Two studies reported the effect of exenatide on total tissue fat reduction compared to the placebo and did not observe a significant difference between the intervention or placebo [28,29]. Other studies did not measure the effects of GLP-1 RAs on total tissue fat reduction. Therefore, NWM was not conducted for this outcome.

### 2.3. Secondary Efficacy Outcomes

#### 2.3.1. Changes in HbA1c, Insulin, and Fasting Plasma Glucose (FPG)

##### HbA1c

Dulaglutide (DMD: −1.39; 95% CI: −2.71, −0.04) was likely to be more effective in HbA1c reduction than the placebo. Liraglutide (DMD: −0.44; 95% CI: −1.13, 0.15), semaglutide (DMD: −0.32; 95% CI: −2.77, 2.96), and exenatide (DMD: −0.05; 95% CI: −0.95, 0.81) were not more effective for HbA1c reduction compared to the placebo. No significant difference was observed when comparing GLP-1 RA to the other GLP-1 RAs (Appendix A). When ranking four GLP-1 RAs, dulaglutide had the highest probability of being ranked first, followed by liraglutide, semaglutide, and exenatide (Table 2).

##### Insulin

Only exenatide and liraglutide were used in these studies. In comparison to the placebo, exenatide (DMD: −2.98; 95% CI −9.45, 3.45) and liraglutide (MD: −3.17; 95% CI −15.6, 8.66) were not associated with decreased insulin levels, as seen in Appendix A. Exenatide had the highest probability of being ranked first, followed by liraglutide (Table 2).

##### FPG

Liraglutide (DMD: −1.87; 95% CI: −3.67, −0.247) was likely to be more effective in FBG reduction than the placebo. Dulaglutide (DMD: −1.96; 95% CI: −4.76, 0.849) and exenatide (DMD: 0.12; 95% CI: −2.77, 2.96) were not effective for FBG reduction compared to the placebo. FPG was not reported in the semaglutide trial. No significant difference was observed when comparing GLP-1 RA to the other GLP-1 RAs (Appendix A). When ranking three GLP-1 RAs, dulaglutide had the highest probability of being ranked first, followed by liraglutide and exenatide (Table 2).

#### 2.3.2. Changes in Systolic and Diastolic Blood Pressure (SBP and DBP)

Compared with the placebo, all three agents failed to demonstrate a significant reduction in SBP (semaglutide: DMD, −1.91; 95% CI, −7.82, 3.66; liraglutide: DMD, −1.23; 95% CI, −5.46, 2.97; exenatide: DMD, −3.29; 95% CI, −7.37, 0.61). Similarly, all three GLP-1 RAs did not significantly reduce DBP versus placebo (semaglutide: DMD, −0.62; 95% CI, −4.95, 3.70; liraglutide: DMD, −0.31; 95% CI, −2.88, 3.26; exenatide: DMD, −1.34; 95% CI, −4.63, 2.12) (Appendix A). SUCRA-based ranking suggested that exenatide was the most likely to reduce SBP and DBP, followed by semaglutide and liraglutide (Table 2).

#### 2.3.3. Changes in Low-Density Lipoproteins (LDLs), Triglycerides (TGs), and Total Cholesterol

Four studies used exenatide and reported the treatment difference in absolute changes in LDLs, TGs, and total cholesterol [26,28,29,30]. Two RCTs used liraglutide and measured treatment difference on ratio to baseline [22,24]. One RCT used semaglutide and reported a significant reduction in percentage changes in LDLs, TGs, and total cholesterol compared to the placebo [27]. Therefore, no pooled data could be synthesized for this outcome.

#### 2.3.4. Quality of Life (QoL)

The effects of semaglutide, liraglutide, and exenatide on scores for the impact of weight on quality of life—Kids (IWQOL–Kids) were measured [24,27,29]. Semaglutide (DMD: 4.27; 95% CI: −2.18, 10.8), liraglutide (DMD: 1.32; 95% CI: −4.59, 7.32), and exenatide (DMD: −4.12; 95% CI: −12.1, 3.75) failed to show a significant QoL improvement compared to the placebo. No significant difference was observed when comparing GLP-1 RA to the other GLP-1 RAs (Appendix A). When ranking three GLP-1 RAs, semaglutide had the highest probability of being ranked first, followed by exenatide and then exenatide (Table 2).

### 2.4. Subgroup Analyses

#### 2.4.1. All Participants with Therapy Duration ≥ 26 Weeks

Semaglutide (DMD: −17.72 kg; 95% CI: −35.74, −0.59) significantly reduced body weight versus placebo; however, other GLP-1 RAs were not associated with significant weight loss. Among all studied GLP-1 RAs, only dulaglutide significantly lowered HbA1c vs. placebo (DMD: −1.41; 95% CI: −2.79, −0.02). Compared to the placebo, semaglutide, liraglutide, exenatide, and dulaglutide were not associated with significant waist circumference, BMI, and BMI z score reductions. No significant difference was found between different GLP-1 RAs regarding the reductions in weight, BMI, BMI z score, and A1c (Appendix A). When ranking four GLP-1 RAs, semaglutide had the highest probability of being ranked first.

#### 2.4.2. Participants with Obesity and All Durations of Therapy

In contrast to the placebo, semaglutide was associated with greater decreases in weight (DMD: −17.7 kg; 95% CI: −22.8, −12.7), waist circumference (DMD: −12.1 cm; 95% CI: −24.6, −0.32), BMI z score (DMD: −1.00; 95% CI: −1.74, −0.22), and A1c (DMD: −0.30; 95% CI: −0.58, −0.002). Exenatide also led to greater weight loss (DMD: −3.47 kg; 95% CI: −5.82, −1.24) versus placebo. Liraglutide, however, was not significantly better than the placebo in all primary outcomes. Moreover, semaglutide was associated with greater weight loss than exenatide (DMD: −14.3 kg; 95% CI: −19.6, −8.68) and liraglutide (DMD: −15.4 kg; 95% CI: −21., −9.79). No significant difference in A1c, BMI z score, or waist circumference reductions was observed when comparing different GLP-1 RAs (Appendix A). When ranking three GLP-1 RAs, semaglutide had the highest probability of being ranked first.

#### 2.4.3. Participants with Obesity and Therapy Duration ≥ 26 Weeks

Semaglutide was associated with greater weight reduction over the placebo (DMD: −17.7 kg; 95% CI: −35.0, −0.66). However, there were no significant differences in BMI, BMI z score, A1c, or waist circumference between semaglutide and the placebo. Neither liraglutide nor exenatide showed significant reductions in weight, BMI, BMI z score, A1c, or waist circumference compared to the placebo. No substantial differences were found between different GLP-1 RAs in terms of these outcomes (Appendix A). When ranking all agents, semaglutide had the highest probability of being ranked first in terms of efficacy for weight reduction.

#### 2.4.4. Participants with Overweight and Diabetes

Only dulaglutide and liraglutide were studied among adolescents with overweight and type 2 diabetes. Neither dulaglutide nor liraglutide demonstrated significant reductions in A1c or BMI z score compared to the placebo. Furthermore, no significant differences were observed between dulaglutide and liraglutide in terms of A1c and BMI z score reductions (Appendix A). Liraglutide ranked first for BMI z score reduction, while dulaglutide ranked first for A1c reduction.

### 2.5. Safety Outcomes

GLP-RAs were not associated with higher odds of causing diarrhea, headache, and abdominal pain compared with the placebo. Liraglutide was noted to cause nausea (OR: 4.31; 95% CI: 1.74, 19.72), vomiting (OR: 6.52; 95% CI: 1.42, 41.31), hypoglycemia (OR: 2.48; 95% CI: 1.13, 6.91), and injection-site reactions (OR: 3.07 * 10^7^; 95% CI: 8.61, 1.02 * 10^28^) than placebo. Furthermore, liraglutide had higher odds of causing an injection-site reaction than semaglutide (OR: 4.85 * 10^7^; 95% CI: 8.15, 2.55 * 10^28^), exenatide (OR: 2.38 * 10^7^; 95% CI: 5.21, 9.14 * 10^27^), and dulaglutide (OR: 3.67 * 10^7^; 95% CI: 8.03, 1.17 * 10^28^). Nonetheless, no significant difference was observed when comparing one GLP-1 RA to other GLP-1 RAs in terms of vomiting, nausea, diarrhea, headache, abdominal pain, and hypoglycemia (Appendix A).

In the rankogram, liraglutide ranked first to cause nausea, vomiting, diarrhea, and injection-site reaction. Dulaglutide ranked first to cause headache, semaglutide first for abdominal pain, and exenatide first for hypoglycemia (Table 3).

## 3. Discussion

This network meta-analysis was based on 11 RCTs, including 953 patients with obesity or overweight who randomly received either a GLP-1 RA or a placebo. Semaglutide exhibited a significantly greater effect in reducing weight, BMI, and BMI z score compared to the placebo. Exenatide also led to notable weight loss when compared to the placebo. Dulaglutide was likely to be more effective in A1c reduction than the placebo. Liraglutide was associated with greater FBG reduction than the placebo. Furthermore, in terms of comparative efficacy among the GLP-1 RAs, semaglutide was associated with greater weight loss, BMI z score reduction, and BMI reduction compared to exenatide. Semaglutide also showed greater weight loss and BMI z score reduction compared to liraglutide and dulaglutide. When ranking all agents, semaglutide had the highest probability of being ranked first for the reductions in actual weight, BMI, BMI z score, and waist circumference.

The higher efficacy of semaglutide relative to other GLP-1RAs has been largely attributed to its distinct molecular properties. Semaglutide possesses a unique structural modification, an acyl group with a stearic acid side chain [31]. This structural enhancement enhances its binding affinity for the GLP-1 receptor and also increases its albumin binding ability in the bloodstream, which helps maintain more sustained drug concentrations. Moreover, the effectiveness of semaglutide in promoting weight loss is believed to be mediated by its interaction with receptors in the central nervous system. This molecular configuration facilitates its penetration into broader regions of the nervous system, thus amplifying its effect on body weight reduction [32].

Similarly, in adults with obesity and without diabetes, semaglutide demonstrated significantly higher odds of achieving more significant reductions in actual weight and BMI compared with the placebo and liraglutide [33,34]. Studies on adults have suggested that the weight loss effects of GLP-1 RAs are associated with decreases in waist circumference, central fat mass, and total body fat [35,36,37]. In adults with overweight and obesity, semaglutide also demonstrated greater weight loss than the placebo and liraglutide [38]. In patients with type 2 diabetes, both liraglutide and exenatide significantly decreased waist circumference versus the placebo [38]. However, in this NWM, none of the four GLP-1 RAs were associated with a significant waist circumference reduction compared to the placebo. The total fat tissue reduction effects of different GLP-1 RAs could not be evaluated in this study due to the lack of relevant data from original studies.

Furthermore, in adults with obesity and without diabetes, a significant reduction in A1c was achieved in patients who received semaglutide compared to those administered the placebo and liraglutide [33,34]. Overall, the effect on A1c reduction was much more significant on children with diabetes or prediabetes than those without diabetes [19]. In this study, similar patterns were observed. Dulaglutide (DMD: −1.39; 95% CI: −2.71, −0.04) was likely to be more effective in A1c reduction than the placebo in children with diabetes. Moreover, in patients with obesity and without diabetes, semaglutide was more effective than the placebo (DMD: −0.3; 95% CI: −0.58, −0.002) in terms of A1c reduction. Similar effects on A1c reduction were observed across different GLP-1 RAs.

In patients with type 2 diabetes, GLP-1 RAs slightly, but significantly, decreased SBP, with only a minimal reduction in DBP [39]. Previous meta-analysis showed GLP-1RAs effectively reduced systolic blood pressure, not diastolic blood pressure, in children and adolescents with obesity or severe obesity [19]. In this study, we did not observe a significant effect of GLP-1 RAs on reducing SBP and DBP among children and adolescents with obesity or overweight. An excess of fat tissue regardless of age and sex has been associated with increased blood pressure [40]; therefore, reductions in fat tissue may result in improvements in blood pressure and lipid profiles [41]. In this NMA, the effect of exenatide on total fat tissue reduction and blood pressure reduction was not significant [28,29]. Since BMI is a poor predictor of fat tissue in children or adolescents [42], it is recommended that total fat tissue reduction should be reported in future studies.

As for the lipid measurements, only a 68-week use of semaglutide and 26-week use of exenatide demonstrated a significant reduction in LDL and total cholesterol [26,27]. A comparison between different GLP-1 RAs could not be conducted in this NMA due to the lack of data from original RCTs. Previous studies found that semaglutide decreased triacylglycerol, LDL, and non-HDL cholesterol levels in patients with diabetes and obesity [43,44].

Vomiting, nausea, dyspepsia, diarrhea, constipation, and abdominal pain are common adverse effects of GLP-1 RA [34]. A previous study showed a significantly increased risk of nausea, but not diarrhea, vomiting, and abdominal pain in adolescents treated with GLP-1 RAs [19]. Liraglutide was most likely to be associated with withdrawal and adverse events compared to other agents in children with obesity [45]. This study found that liraglutide was more likely to cause nausea, vomiting, hypoglycemia, and injection-site reactions than the placebo. All four GLP-1 RAs were not associated with higher odds of causing diarrhea, headache, or abdominal pain compared with the placebo. We also observed that liraglutide had higher odds of causing injection-site reactions than semaglutide, exenatide, and dulaglutide. Nonetheless, no significant difference was observed when comparing one GLP-1 RA to other GLP-1 RAs in terms of other adverse events.

Subgroup analyses showed that the pooled results of patients who received treatment for more than 26 weeks were not significantly better than the placebo, or the effect size was smaller compared to that among patients, regardless of treatment duration. Two RCTs also demonstrated that the treatment effects were more pronounced with short-term therapy than with extended durations [24,29]. This may have occurred with a rebound effect of the intervention [46]. It is possible that the relatively lower executive functioning of youths and adolescents may make weight loss maintenance particularly challenging [47]. Therefore, an engagement in lifestyle modification interventions and the use of medications should be continued indefinitely if treatments are well tolerated [48]. Face-to-face physical activity combined with dietary intervention and family-based treatment have demonstrated long-term benefits in this population and should be advocated [49,50].

The investigation of combinations involving GLP-1, gastric inhibitory polypeptide (GIP), islet amyloid polypeptide, glucagon, and leptin has revealed their potential to enhance weight loss in adults with obesity and type 2 diabetes [51]. GIP directly affects the central nervous system by suppressing food intake and amplifying the anorexigenic properties of GLP-1 [52]. Tirzepatide, a promising agent that exhibits GLP-1/GIP co-agonism, significantly reduced body weight across all doses in adults with obesity [53,54]. Furthermore, amylin, as a potential enhancer of the effects of GLP-1 on satiation signals, may synergistically interact with GLP-1 and augment the efficacy of leptin [55]. Clinical studies have demonstrated that the co-administration of leptin and pramlintide yielded greater weight loss than either agent alone [56]. Future clinical studies should consider these combinations for the pediatric population.

### Strengths and Limitations

The present review demonstrated several notable strengths. Firstly, the researchers conducted an extensive and systematic search and identified a comprehensive range of variables to reflect the effect of GLP-1 RAs on individuals with obesity and overweight, including reductions in actual weight, BMI, BMI z score, and waist circumference. Furthermore, their effects on blood pressure, lipids levels, A1c, and QOL were also analyzed to broaden the scope of this NMA. This study compared all four GLP-1 RAs, including the one recently approved by the FDA, thereby enhancing its inclusiveness and comprehensiveness of coverage. The assessment of the risk of bias was carried out using the Cochrane Risk of Bias 2 tool, which added credibility and robustness to the evaluation process. Notably, the authors employed network meta-analysis and performed subgroup analyses, enabling a more nuanced and insightful interpretation of the data.

However, there are limitations to consider. Specifically, the absence of certain information, such as means and standard deviations in the primary studies, was a concern. Our attempts to obtain this information from the authors were unsuccessful. The duration of the trials included in our analysis limits our ability to fully assess the long-term implications of GLP-1 receptor agonist use in pediatric populations. Furthermore, the review could not account for the variability in medication dosages in various trials, which could have potentially influenced the outcomes. Additionally, a direct comparison of GLP-1 RAs could not be performed due to a lack of head-to-head comparative efficacy and safety studies. Lastly, due to the limited number of studies available, the publication bias could not be assessed, and thus these findings should be interpreted as indicative rather than definitive. Future research should aim to address these gaps by including longer-term studies, uniform dosing strategies, and direct comparisons of these medications to evaluate the efficacy and safety of these treatments more comprehensively.

## 4. Materials and Methods

### 4.1. Search Strategy

Searches on databases, including PubMed, Scopus, and Embase, were performed in March 2023 and updated in June 2024 without the filters of date or language. Key words included “obesity” or “overweight” or “obese” AND “child” or “children” or “pediatric” or “youth” or “adolescents” AND “glucagon-like peptide-1 receptor agonists” or “GLP-1RA” or “exenatide” or “dulaglutide” or “semaglutide” or “liraglutide” AND “randomized” or “randomised” or “random” or “RCT” or “randomized controlled trial” or “trial”. The search strategy is provided in the Appendix A as well. After conducting a preliminary review of the relevant literature, we registered this project on the PROSPERO website (ID: CRD42023411992). We also reviewed recently published reviews and meta-analyses relevant to our topic to ensure comprehensive coverage of the existing studies. All subsequent analyses were carried out according to the registered protocol.

### 4.2. Study Selection

The study selection process was carried out using the systematic review web-based tool COVIDENCE (covidence.org). We included parallel or crossover designed RCTs that met the following criteria: (1) the population consisted of patients under 18 years of age, diagnosed with obesity or overweight; (2) the intervention was GLP-1 RA; (3) the comparison was either a placebo or another GLP-1 RA; and (4) the outcomes were any predefined clinical measures, such as weight reduction, glycemic control, or other health-related impacts. This approach aligned with the population, intervention, comparison, and outcome (PICO) framework, ensuring a targeted and methodical analysis. Overweight was defined as a BMI between the 85th and 95th percentiles for age and sex, obesity as BMI ≥ 30 kg/m^2^, or in the 95th or higher percentiles, and severe obesity as BMI ≥ 1.2 times the 95th percentile or BMI ≥ 35 kg/m^2^ [16]. Participants with type 1 diabetes, Prader–Willi syndrome, hypothyroidism, or a history of eating disorders were excluded. Two authors (L.L. and H.T.) independently reviewed all the articles by reading titles and abstracts following the literature search. Each full article was subsequently assessed for eligibility.

### 4.3. Outcome Measures

The primary efficacy outcomes in this network meta-analysis (NMA) included: (1) mean change in actual weight; (2) mean changes in BMI and/or BMI z score; and (3) changes in waist circumference and body fat composition. The secondary efficacy outcomes included changes in (1) hemoglobin A1c (HbA1c), fasting plasma glucose (FPG), and insulin; (2) triglycerides (TGs), low-density lipoprotein (LDL), and total cholesterol (TC); (3) systolic and diastolic blood pressure (SBP, DBP); and (4) the impact of weight on quality of life—kids (IWQOL-Kids). All efficacy outcomes were reported as difference-in-mean-difference (DMD) for the comparison with different GLP-1 RAs or placebo. Safety outcomes were frequently reported adverse events, including the proportion of patients with nausea, vomiting, diarrhea, abdominal pain, headache, injection-site reaction, and hypoglycemia as defined by the American Diabetes Association [57].

### 4.4. Data Extraction and Risk of Bias Assessment

Baseline characteristics, sample size, study design, intervention and control groups, duration of therapy, and predefined outcomes were extracted from each of the included studies by two authors (Y.S. and N.G.) and reviewed by L.L. Two reviewers (L.L. and A.W.) evaluated the risk of bias of included studies independently using the Cochrane Collaboration’s tool for randomized trials [58].

### 4.5. Data Analyses

In the network diagram, each node represented a GLP-1 RA or placebo, with the node size indicating the sample size and the line thickness reflecting the number of studies included. Using a Bayesian approach, a random-effects NMA was performed. For pairwise comparisons, odds ratio (OR) or DMD with 95% confidence intervals (CIs) was used for binary and continuous variables. The heterogeneity was determined by I^2^, where a value >50% represented moderate heterogeneity. Inconsistency was not assessed in this study because no head-to-head trials comparing two different GLP-1 Ras were conducted. To assess the likelihood of comparative efficacy and safety of GLP-1, surface under the cumulative ranking probabilities (SUCRAs) were calculated. A lower SUCRA value (ranging from 0 to 1) suggested reduced efficacy, while a higher value indicated a greater likelihood of experiencing adverse events. Publication bias could not be determined using the Egger test due to the limited number of published studies included. The network meta-analysis was conducted using R software (version 4.1.2) with the netmeta package. This NMA adhered to the preferred reporting items for systematic reviews and meta-analyses involving NMA (PRISMA-NMA) [59].

## 5. Conclusions

In children and adolescents with obesity, semaglutide appeared to be the most effective and safe option among four GLP-1 RAs. While this evidence could potentially inform clinical decisions regarding the management of obesity or overweight in pediatric populations, it is important to interpret these results with caution due to the limited number of published studies. Therefore, additional head-to-head trials comparing GLP-1 RAs are essential to substantiate these preliminary findings and provide more definitive guidance for healthcare providers about their optimal use.

## Figures and Tables

**Figure 1 pharmaceuticals-17-00828-f001:**
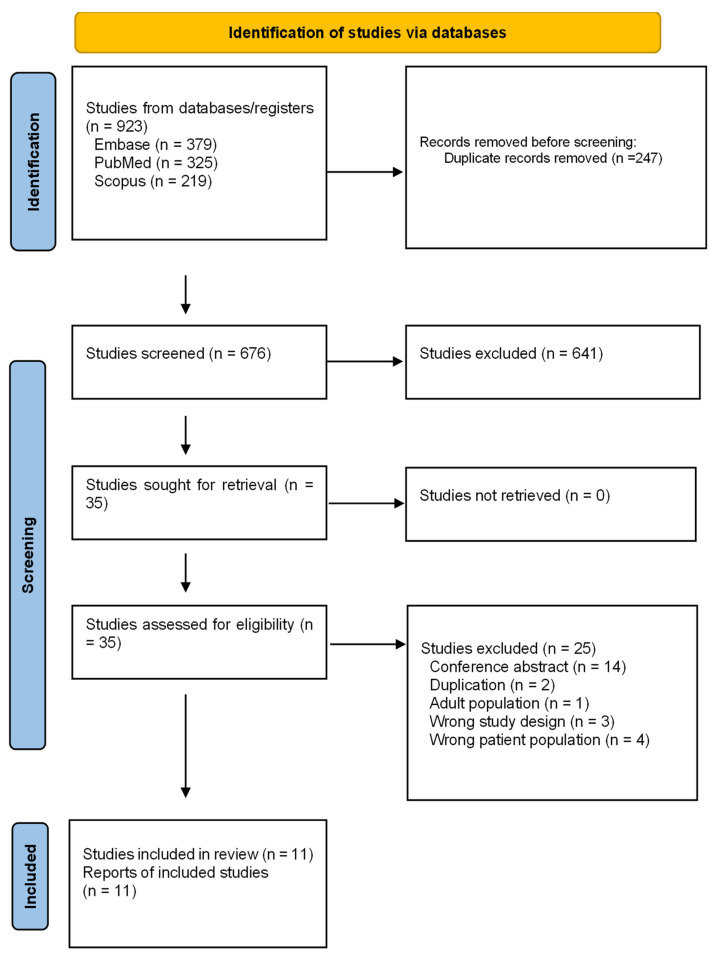
Study selection process.

**Figure 2 pharmaceuticals-17-00828-f002:**
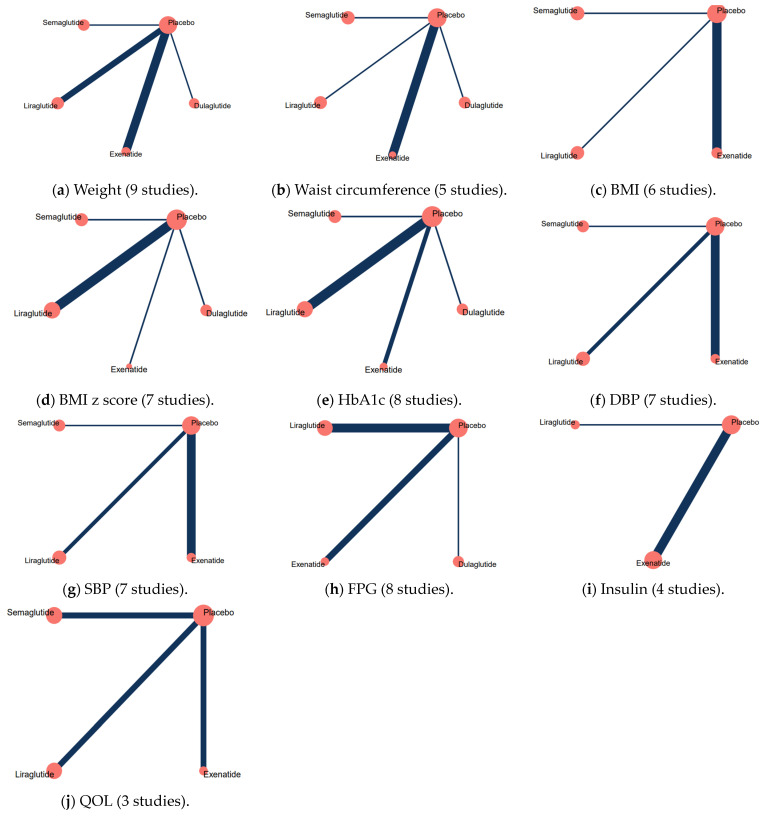
Network graphs of each outcome of the main analysis: (**a**) weight, (**b**) waist circumference, (**c**) BMI, (**d**) BMI z score, (**e**) HbA1c, (**f**) DBP, (**g**) SBP, (**h**) FPG, (**i**) insulin, and (**j**) QOL. Notes: 
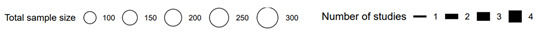
. Abbreviations: BMI, body mass index; HbA1c, hemoglobin A1C; DBP, diastolic blood pressure; SBP, systolic blood pressure; FPG, fasting plasma glucose; QOL, quality of life.

**Figure 3 pharmaceuticals-17-00828-f003:**
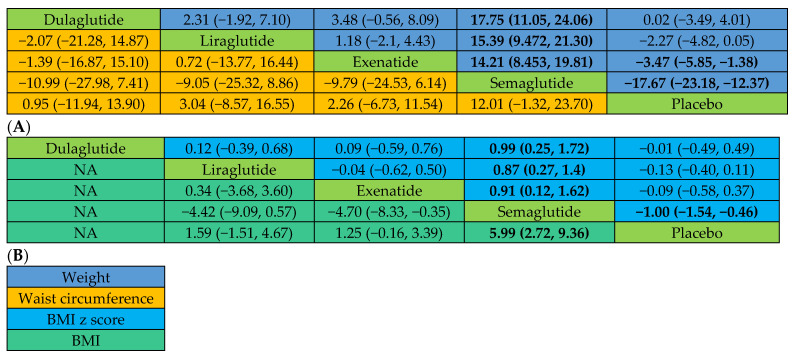
League tables of main outcome analyses. (**A**) Actual body weight and waist circumference. (**B**) BMI and BMI z score. The league tables show the relative effects of each medication (the treatment on the column to the treatment of the row). The relative effects are measured as a difference in mean difference (DMD) with a 95% CI for mean change in actual body weight (kg), waist circumference (cm), BMI (kg/m^2^), and BMI z score. Bold indicates statistical significance. NA: comparison not available.

**Table 1 pharmaceuticals-17-00828-t001:** Characteristics of included studies and baseline population characteristics of included studies.

First Author, Year	Country	Design	Population (Number of Patients)	Age (Years)	Male, (%)	Weight (Kg),	BMI (Kg/m^2^),	BMI z Score	Intervention	Dose Regimen	Control
Kelly, 2012 [30]	United States	Crossover RCT	Extreme obesity (N = 11)	12.7 (2.1)	18.2	93.8 (20.6)	36.7 (4.8)	NI	13 weeks of exenatide	Initiated at 5 mcg twice daily. After 1 month, up-titrated to 10 mcg twice daily for the remaining 2 months.	Volume-matched placebo
Kelly, 2013 [28]	United States	Randomized, double-blind, placebo controlled, multicenter clinical trial	Severe obesity (N = 26)	15.2 (1.8)	38.5	124 (19.3)	42.5 (6.81)	NI	13 weeks of exenatide	Initiated at a dose of 5 mcg twice daily. Up-titrated to 10 mcg twice daily after 1 month for the remainder of the trial. If the 10 mcg dose was not tolerated, the dose was reduced to 5 mcg.	Volume-matched placebo pen
Weghuber, 2020 [26]	Sweden; Austria	Parallel, double-blinded, randomized, placebo controlled two-arm trial	Obesity (N = 44)	14.4 (2.3)	50	104.4 (21.7)	36.1 (4.9)	3.2 (0.5)	26 weeks of exenatide	Initiated at a dose of 0.6 mg, and was increased by 0.6 mg/week to a maximum of 3 mg/day.	Volume-matched placebo pen
Fox, 2022 [29]	United States	Randomized, double-blind, placebo-controlled trial	Severe obesity (N = 66)	16 (1.5)	53	108.5 (17.6)	36.9 (4.4)	NI	52 weeks of exenatide	Initiated at 0.3 mg daily for first week and increased weekly thereafter to 0.6 mg, 0.9 mg, 1.2 mg, and 1.8 mg.	Matching placebo devices
Klein, 2014 [21]	Belgium Slovenia United Kingdom US	Double-blind, randomized, placebo-controlled, parallel group trial	Overweight and type 2 diabetes (N = 21)	15 (NI)	33.3	113 (NI)	40 (NI)	3.4 (0.7)	5 weeks of liraglutide	Escalated from 0.3 to 1.2 mg in weekly increments of 0.3 mg and then followed with 0.6 mg weekly increments to a maximum dose of 3 mg or maximum tolerated dose.	Volume-matched placebo pen
Danne, 2017 [23]	Germany	Randomized, double-blind, parallel-group, placebo-controlled trial	Obesity (N = 21)	14.9 (1.3)	33.3	105.5 (20.5)	36 (4.0)	3.2 (0.59)	5 weeks of liraglutide	Initiated at the 0.75 mg dose for the first 4 weeks and then escalated to the 1.5 mg dose if the participant did not have unacceptable side effects with the lower dose.	Volume-matched placebo pen
Mastrandrea, 2019 [25]	United States	Randomized, double-blind, placebo-controlled trial	Obesity (N = 24)	9.9 (1.1)	37.5	71.5 (15.4)	NI	3.9 (0.9)	7 to 13 weeks of liraglutide	Weekly subcutaneous injections of exenatide; 2 mg	Equal volume of placebo
Tamborlane, 2019 [22]	84 sites in 25 countries	Randomized, parallel-group, placebo-controlled trial	Overweight and type 2 diabetes (N = 134)	14.6 (1.7)	38.1	91.5 (26.8)	33.90 (9.25)	2.94 (1.30)	52 weeks liraglutide	liraglutide was initiated at a dose of 0.6 mg per day and was escalated in both groups in increments of approximately 0.6 mg each week over the course of 2 to 3 weeks.	Visually identical prefilled pen injectors
Kelly, 2020 [24]	Belgium	Randomized, double-blind, placebo-controlled, phase 3 trial	Obesity (N = 251)	14.6 (1.6)	40.6	100.75 (20.95)	35.55 (5.44)	3.17 (0.71)	56 weeks of liraglutide	Initiated and maintained at a dose of 2 mg weekly.	Volume-matched placebo pen
Arslanian, 2022 [20]	Nine countries	Phase 3, randomized, placebo-controlled, parallel-group, superiority trial	Overweight and type 2 diabetes (N = 154)	14.5 (2.0)	29	90.5 (26.5)	34.1 (8.8)	2.94 (NI)	26 weeks of dulaglutide	Initiated at a dose of 0.6 mg daily for 1 week; increased weekly thereafter until the maximum tolerated dose or 3 mg daily.	Visually identical, single-use, single-dose pen devices
Weghuber, 2022 [27]	Austria, Belgium, Croatia, Ireland, Mexico, Russian Federation, United Kingdom, United States	Multinational, double-blind, parallel-group, randomized, placebo-controlled, phase 3a trial	Obesity (N = 201)	15.4 (1.6)	38	107.5 (24.5)	37.0 (6.4)	3.31 (0.86)	68 weeks of semaglutide	Initiated at a dose of 0.25 mg once weekly for the first 4 weeks, followed by escalation every 4 weeks to 0.5, 1.0, 1.7, and 2.4 mg.	Matching placebo

Notes: Age, weight, BMI, and BMI z score were presented as mean (SD). SD, standard deviation; BMI, body mass index, NI, no information.

**Table 2 pharmaceuticals-17-00828-t002:** Treatment ranking tables of SUCRA values for efficacy outcomes.

	Dulaglutide	Exenatide	Liraglutide	Semaglutide	Placebo
Weight	15.47%	68.65%	52.41%	**99.95%**	13.52%
BMI z score	27.45%	46.04%	57.75%	**98.94%**	19.82%
BMI	NI	44.97%	53.06%	**98.03%**	3.95%
Waist circumference	34.04%	48.26%	54.57%	**93.00%**	20.13%
HbA1c	**93.74%**	27.03%	61.63%	48.48%	19.13%
SBP	NI	**81.48%**	45.44%	56.64%	16.44%
DBP	NI	**72.40%**	32.38%	55.98%	39.24%
FPG	**78.94%**	23.32%	77.88%	NI	19.86%
Insulin	NI	**67.03%**	61.55%	NI	21.43%
QOL	NI	11.12%	59.44%	**88.48%**	40.97%

Abbreviations: BMI, body mass index; HbA1c, hemoglobin A1C; DBP, diastolic blood pressure; SBP, systolic blood pressure; FPG, fasting plasma glucose; QOL, quality of life; NI, no information. A higher value indicates a greater benefit (larger decrease in the outcome of interest) of a certain intervention. Bolded are training modalities identified as the best for the given outcome.

**Table 3 pharmaceuticals-17-00828-t003:** Treatment ranking tables of SUCRA values for safety outcomes.

	Dulaglutide	Exenatide	Liraglutide	Semaglutide	Placebo
Nausea	44.72%	56.95%	**76.82%**	63.40%	8.11%
Vomiting	58.51%	62.90%	**66.37%**	55.23%	6.99%
Diarrhea	54.48%	39.21%	**81.48%**	43.45%	31.39%
Abdominal pain	35.31%	29.25%	73.54%	**77.28%**	34.63%
Headache	**69.80%**	48.79%	45.39%	45.13%	40.89%
Injection-site reaction	34.96%	49.81%	**99.56%**	27.99%	37.69%
Hypoglycemia	19.42%	**74.93%**	73.00%	NI	32.65%

A higher value indicates a greater chance of adverse reactions to certain interventions. Bolded are training modalities identified as the worst for the given outcome. NI: No information.

## Data Availability

The original contributions presented in the study are included in the article/Appendix A; further inquiries can be directed to the corresponding author.

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
