# Peer review of "Comparative Efficacy and Safety of Glucagon-like Peptide-1 Receptor Agonists in Children and Adolescents with Obesity or Overweight: A Systematic Review and Network Meta-Analysis"

_pharmaceuticals, 2024, doi:10.3390/ph17070828_

Round 1
Reviewer 1 Report
Comments and Suggestions for Authors
Obesity in children and adolescents is currently an emerging health problem worldwide. It is associated with many health complications, as well as many chronic diseases such as cardiovascular diseases, diabetes, hypertension, non-alcoholic fatty liver diseases, polycystic ovarian syndrome, and sleep apnea. Glucagon-like peptide-1 receptor agonists are currently approved for use in treating obesity in the pediatric population. Their efficacy and safety profile are of great interest to the research community. The article entitled “Comparative Safety and Efficacy of Glucagon-Like Peptide-1 Receptor Agonists in Children and Adolescents with Obesity or Overweight: A Systematic Review and Network Meta-Analysis” aimed to review the comparative efficacy and safety of the four approved GLP-1 Ras in pediatric population with obesity or overweight with type 2 diabetes. The manuscript is very well written and can be of high interest to the research community and clinicians.
Some comments for the authors to improve the manuscript:
1. Figure 2 – the text is not readable, I suggest modifying the figures accordingly.
2. The abbreviations should be defined at the first use in the text, ex. FPG line 251, BP line 273, LDL, TG line 283, etc
3. In the material and methods section a schematic representation of study selection can be beneficial for readers.
Author Response
Reviewer 1:
Obesity in children and adolescents is currently an emerging health problem worldwide. It is associated with many health complications, as well as many chronic diseases such as cardiovascular diseases, diabetes, hypertension, non-alcoholic fatty liver diseases, polycystic ovarian syndrome, and sleep apnea. Glucagon-like peptide-1 receptor agonists are currently approved for use in treating obesity in the pediatric population. Their efficacy and safety profile are of great interest to the research community. The article entitled “Comparative Safety and Efficacy of Glucagon-Like Peptide-1 Receptor Agonists in Children and Adolescents with Obesity or Overweight: A Systematic Review and Network Meta-Analysis” aimed to review the comparative efficacy and safety of the four approved GLP-1 Ras in pediatric population with obesity or overweight with type 2 diabetes. The manuscript is very well written and can be of high interest to the research community and clinicians.
Some comments for the authors to improve the manuscript:
- Figure 2 – the text is not readable; I suggest modifying the figures accordingly.
We appreciate your comments about the readability of the text in Figure 2 and have therefore modified it accordingly. The text size has been increased to ensure that all elements within the figure are clear and readable (Line 169).
- The abbreviations should be defined at the first use in the text, ex. FPG line 251, BP line 273, LDL, TG line 283, etc
We apologize for the oversight in our initial submission concerning the abbreviations. We have defined all abbreviations at their first use within the text: FPG (fasting plasma glucose) on line 261, BP (blood pressure) on line 283, LDL (low-density lipoprotein) and TG (triglycerides) on line 291.
- In the material and methods section a schematic representation of study selection can be beneficial for readers.
Thank you for your suggestion to include a schematic diagram of the study selection process. To adhere to PRISMA guidelines, we reported it in the Results section, not the Methods section, as outlined on Line 96 of the manuscript.

Reviewer 2 Report
Comments and Suggestions for Authors
The review written on GLP-1 antagonists in children and adolescents is relevant, however, several shortcomings need to be resolved.
The last statement of the abstract states that “obesity or overweight plus diabetes”. Were the authors looking for literature in which the intended use of GLP-1 antagonist was only against obesity/overweight or they were also looking for their concurrent use for diabetes?
One major limiting factor for this article is the inclusion of just 11 RCTs for four GLP-1 antagonists. It is difficult to conclude a reliable outcome based on merely 11 studies.
I could not find the classification of studies based on the type of GLP-1 A used.
Figure 1 is more suitable for the methods section. Also, records with zero deletion can be omitted like Records marked as ineligible by automation tools (n =0).
The overall sample size of the selected studies is also less, only four studies have over 100 samples. It is not clear about those studies about the type of GLP-1 A they investigated.
I recommend authors to give a detailed description of the type of GLP-1 A used by each study.
A study by Mastrandrea, 2018 has no information about BMI, why this study was still chosen? weight check was the primary measure of this article.
The studies selected were only between 2017 to 2022, what is the reason for this time?
This conclusion needs to be cautiously re-written considering that the inference drawn is based on very few studies.
“This finding can guide the decisions of healthcare 527 providers when considering a GLP-1 RA for the treatment of obesity or overweight with 528 diabetes among pediatric patients”.
Author Response
Reviewer 2:
The review written on GLP-1 antagonists in children and adolescents is relevant, however, several shortcomings need to be resolved.
- The last statement of the abstract states that “obesity or overweight plus diabetes”. Were the authors looking for literature in which the intended use of GLP-1 antagonist was only against obesity/overweight, or they were also looking for their concurrent use for diabetes?
To clarify, our study investigated the use of GLP-1 RAs in two distinct patient groups: those with obesity, and those who were overweight and concurrently diagnosed with diabetes.
- One major limiting factor for this article is the inclusion of just 11 RCTs for four GLP-1 antagonists. It is difficult to conclude a reliable outcome based on merely 11 studies.
We acknowledge your concern regarding the limited number of RCTs included in our analysis. While it is challenging to draw definitive conclusions from 11 studies, these studies were rigorously selected based on our inclusion criteria to ensure the highest level of scientific integrity and relevance. We have emphasized this in the limitations paragraph in the Discussion section that these findings should be interpreted as indicative rather than definitive, and we have strongly advocated for more extensive research to further elucidate these outcomes (Lines 491 to 493).
- I could not find the classification of studies based on the type of GLP-1 RA used.
Thank you for your suggestion. We have categorized the studies based on the specific GLP-1 RA. This classification is detailed in Table 1 and described in the manuscript on Lines 92-95. We hope this clarifies the categorization for our readers.
- Figure 1 is more suitable for the methods section. Also, records with zero deletion can be omitted like Records marked as ineligible by automation tools (n =0).
We appreciate your suggestions regarding Figure 1. We have removed the records marked with zero deletions (n=0) to streamline the presentation. Regarding the placement of Figure 1, we have adhered to the PRISMA guidelines, which recommend reporting flow diagrams in the Results section. Therefore, Figure 1 has been appropriately included in the Results section of our manuscript, as per these guidelines.
- The overall sample size of the selected studies is also less, only four studies have over 100 samples. It is not clear about those studies about the type of GLP-1 RAs they investigated. I recommend authors to give a detailed description of the type of GLP-1 RAs used by each study.
Thank you for highlighting the need for clearer information regarding the types of GLP-1 RAs investigated in the studies included in our analysis. We apologize for any confusion caused by the initial presentation of this data. To address this, we have moved the detailed descriptions of the GLP-1 RAs used in each study from Supplementary Table 1 to the main text in Table 1. This revision ensures that the information is more accessible and clearer to our readers.
- A study by Mastrandrea, 2018 has no information about BMI, why this study was still chosen? weight check was the primary measure of this article.
Thank you for your query regarding the inclusion of the study by Mastrandrea, 2018, which lacks BMI data. While BMI information was not provided, this study was included in our network meta-analysis (NMA) because it reported on other primary efficacy outcomes relevant to our analysis, such as mean change in actual body weight and changes in waist circumference and body fat composition.
- The studies selected were only between 2017 to 2022, what is the reason for this time?
Thank you for your comments. We conducted a comprehensive database search up to March 2023 to identify RCTs addressing childhood obesity when we finalized the manuscript. Upon a subsequent update of the search in June 2024, however, no new RCTs were identified. This detail has been provided in Lines 496 to 497 of the manuscript.
- This conclusion needs to be cautiously re-written considering that the inference drawn is based on very few studies. “This finding can guide the decisions of healthcare 527 providers when considering a GLP-1 RA for the treatment of obesity or overweight with 528 diabetes among pediatric patients”.
Thank you for your constructive feedback. We acknowledge the concern regarding the limited number of studies and the implications for our conclusions. We have revised the Conclusion to reflect a more cautious tone, emphasizing the preliminary nature of our findings and the need for further research. Please see the updated text in Lines 558 to 562 of the manuscript.

Reviewer 3 Report
Comments and Suggestions for Authors
I attached a document containing my review report.

Minor English editing is needed.
Author Response
Reviewer 3:
OVERALL
With pleasure, I read the paper titled “Comparative Safety and Efficacy of Glucagon-Like Peptide-1 Receptor Agonists in Children and Adolescents with Obesity or Overweight:”. The topic is clinically relevant to practice, and of importance to the readers of the Pharmaceuticals. The introduction section was detailed enough to provide the reader with the needful background information. However, the authors to carefully conclude the section with some proposed hypotheses of the research. The methods section was detailed too, however, some MAJOR and MINOR edits are needed for complete reporting in line with PRISMA guidelines as outlined below. Again, the network meta-analysis of various primary and secondary efficacy and safety endpoints is a key strength of this manuscript. The results section was detailed and informative. The discussion section is well written. However, it needs to provide some elaboration on mechanistic, molecular or pharmacological aspects on semaglutide being more effective. Also, comparison with previously published meta-analysis (Ryan 2021, reference 18) should be discussed. The research had some unavoidable limitations, all of which had been explicitly acknowledged, but additional limitations should be elaborated. The conclusion is line with the presented results. ALL IN ALL, this manuscript is clinically relevant. Some comments are provided below.
ABSTRACT
- Please spell out all abbreviations upon first encounter.
Thank you for pointing it out. We have now revised the abstract to ensure that all abbreviations are spelled out upon their first appearance.
- Please mention the dates of database search (from when to when).
Thank you for your comments. We have now included this information in line 23 of the manuscript to provide full transparency and context for our review process.
INTRODUCTION
- Please briefly mention how liraglutide, semaglutide, dulaglutide, and exenatide differ from each other.
We have added a brief comparison of liraglutide, semaglutide, dulaglutide, and exenatide in our manuscript. Please refer to lines 64-65 for the information.
- The authors may need to conclude the section with some proposed hypotheses.
We have addressed this suggestion by adding hypotheses in the Introduction section of our manuscript. Please see lines 76-77.
METHODS
- The last date of literature search appears somehow outdated (March 2023). I wonder if new RCTs have been published since then. I encourage the authors to re-search the databases quickly and check if new RCTs have been published and not included in their report.
We appreciate your suggestion and have conducted an updated search of the databases in June 2024. We can confirm that no new randomized controlled trials have been published since our last search in March 2023. This has been updated in the manuscript to reflect the most current information (Line 497).
- According to PRISMA guidelines, the specific (exact) literature search strategy for at least one database (e.g., PubMed) should be provided in the main text and/or supplementary file for reproducibility. Also, please include the key words used in literature search.
We have adhered to the PRISMA guidelines by providing our literature search strategy in the main text and supplementary materials. This includes the exact search terms used in PubMed as an example: ('GLP-1 receptor agonists' or 'GLP-1 RA*' or 'semaglutide' or 'liraglutide' or 'exenatide' or 'dulaglutide') and ('adolescent' or 'children' or 'child' or 'pediatric' or 'youth*') and ('obesity' or 'obese' or 'overweight') and ('RCT' or 'randomized' or 'random' or 'trial*') (Lines 497 to 502).
- Have you searched the grey literature for additional studies that could have been missed?
We did not include grey literature in our search. Instead, we reviewed published RCTs as well as reviews and meta-analyses relevant to our topic to ensure comprehensive coverage of existing studies. Based on our inclusion and exclusion criteria, no additional studies from these sources were identified as relevant for inclusion in our analysis (Line 503).
- The authors claimed database search was done without filters of date or language. However, one of the inclusion criteria was RCTs written in English or Chinese. Please resolve this conflict.
We have resolved the discrepancy in the manuscript (Line 512).
- The authors claimed study selection was done through Covidence, which is acceptable. But this conflicts with the claim that “two authors (L.L. and H.T.) independently reviewed all articles by reading titles and abstracts after the literature search. Each full article was assessed for eligibility.”
Thank you for your comment regarding our use of Covidence. Covidence allows each reviewer to independently screen titles and abstracts and later assess full texts for eligibility. This feature of Covidence ensures that our review process is both systematic and independent, allowing reviewers L.L. and H.T. to independently evaluate each article while maintaining the integrity and transparency of the study selection process.
- The inclusion criteria should be provided following the standard evidence-based PICO or PICOS method.
Thank you for your suggestion. We have revised the inclusion criteria to align with the standard PICO format, which is detailed in lines 510 to 515 of the manuscript.
- Did the authors perform any evaluation of the quality of the included studies?
We evaluated the quality of the included studies, and the details of this evaluation are described in lines 160 to 164 of the manuscript.
- Please indicate why the random-effects model was used. Was heterogeneity expected a priori?
Thank you for your inquiry about the choice of model. We opted for a random-effects model due to the expected heterogeneity among the studies, stemming from variations in geographic locations, healthcare settings, medication dosages, and therapy durations. These factors collectively contribute to differences in study outcomes, making the random effects model a more appropriate choice for our analysis. It allows for a more conservative estimate, enhancing the reliability of our findings that semaglutide was more effective, even in a model that accounted for potential variability.
RESULTS
- In table 1, please provide the country of publication.
We have updated Table 1 to include the country of publication for each study.
DISCUSSION
- The second paragraph is pretty similar to results section, and it should be revised or deleted.
Thank you for your comment. To avoid redundancy and maintain the clarity of the manuscript, we have removed this paragraph.
- Please discuss from a molecular/pharmacological perspective why semaglutide was more effective than the other GLP-1 RA agents.
We have elaborated on this aspect in the manuscript from lines 378 to 386, providing a detailed discussion of the unique properties of semaglutide that could contribute to its greater efficacy.
GENERAL
- The manuscript needs some polishing for English language and editing.
Thank you for your feedback regarding the language quality of our manuscript. We have carefully revised the text to enhance readability.
- In consideration of the large number of figures, I encourage the authors to double-check them and make sure everything looks fine
Thank you for your recommendation. We have thoroughly reviewed all figures to ensure accuracy, clarity, and proper presentation. All necessary revisions have been made to enhance visual representation and support the data discussed within the manuscript.

Reviewer 4 Report
Comments and Suggestions for Authors
1. These authors have analyzed 11 articles which reported results from randomized trials of children and adolescents who received glucagon-like peptide-1 agonists. The primary outcomes included changes in body weight, in BMI, and in waist circumference. Safety outcomes included nausea, vomiting, diarrhea, abdominal pain, injection site reactions, and hypoglycemia. They describe this study as a network meta-analysis. However, none of the drugs were compared in head-to-head trials.
2. Semaglutide had the largest effect in reducing weight, BMI, and BMI Z-score and had better weight loss effects than the other 3 drugs in these various trials. Side effects with all 4 drugs were similar to placebo. Overall, they concluded that semaglutide was the most effective drug and was a safer option for children and adolescents.
3. The longest of trial lasted 68 weeks. Consequently, it is difficult to know the long-term benefits and hazards associated with this approach to weight loss. In addition, the authors note several other important limitations in their study and the data available for analysis. First, they were unable to account for variation in medication dosages. Second, there was no direct comparison of these 4 drugs. Last, they could not analyze the data for publication bias, but it seems likely that studying new drugs in an important medical problem for large populations of individuals will result in publication.
Author Response
- These authors have analyzed 11 articles which reported results from randomized trials of children and adolescents who received glucagon-like peptide-1 agonists. The primary outcomes included changes in body weight, in BMI, and in waist circumference. Safety outcomes included nausea, vomiting, diarrhea, abdominal pain, injection site reactions, and hypoglycemia. They describe this study as a network meta-analysis. However, none of the drugs were compared in head-to-head trials.
Thank you for your comment. We acknowledge the limitation as noted in the Discussion section regarding the lack of head-to-head trials in our network meta-analysis. Future studies would benefit from direct comparisons between GLP-1 receptor agonists to provide stronger evidence on their relative efficacy and safety. We have noted this as an important direction for upcoming research in our Discussion and Conclusions section (Line 567).
- Semaglutide had the largest effect in reducing weight, BMI, and BMI Z-score and had better weight loss effects than the other 3 drugs in these various trials. Side effects with all 4 drugs were similar to placebo. Overall, they concluded that semaglutide was the most effective drug and was a safer option for children and adolescents.
Thank you for summarizing the key findings of our analysis. Semaglutide had better efficacy in managing weight and BMI compared to the other evaluated GLP-1 receptor agonists. As for the safety profile, our analysis indicated that the side effects were indeed similar to placebo, which reinforces the potential of semaglutide as a viable option for pediatric obesity management. Your observation was helpful to emphasize the significant implications of our results for clinical practice.
- The longest of trial lasted 68 weeks. Consequently, it is difficult to know the long-term benefits and hazards associated with this approach to weight loss. In addition, the authors note several other important limitations in their study and the data available for analysis. First, they were unable to account for variation in medication dosages. Second, there was no direct comparison of these 4 drugs. Last, they could not analyze the data for publication bias, but it seems likely that studying new drugs in an important medical problem for large populations of individuals will result in publication.
Thank you for highlighting the limitations noted in our study. The duration of the trials included in our analysis limits our ability to fully assess the long-term implications of GLP-1 receptor agonist use in pediatric populations. Additionally, the variability in medication dosages and the lack of direct head-to-head comparisons between the drugs are significant gaps that could affect the robustness of our conclusions. We acknowledge the possibility of publication bias, particularly given the high interest in new therapeutic options for a prevalent issue like pediatric obesity. Future research should aim to address these gaps by including longer-term studies, uniform dosing strategies, and direct comparisons of these medications to build a more comprehensive evidence base (Lines 484 to 497).

Round 2
Reviewer 1 Report
Comments and Suggestions for Authors
The authors addressed all my concerns, so in its current form, the manuscript is ready for acceptance.